# Synthesis of aryldifluoromethyl aryl ethers via nickel-catalyzed suzuki cross-coupling between aryloxydifluoromethyl bromides and boronic acids

Heng Lu[1,2], Ruo-Xuan Xiao[1,2], Chang-Yun Shi[1], Zi-Lan Song[1], Hou-Wen Lin[1] & Ao Zhang [1✉]

As a unique organofluorine fragment, gem-difluoromethylated motifs have received widespread attention. Here, a convenient and efficient synthesis of aryldifluoromethyl aryl ethers (ArCF$_2$OAr') was established via Nickel-catalyzed aryloxydifluoromethylation with arylboronic acids. This approach features easily accessible starting materials, good tolerance of functionalities, and mild reaction conditions. Diverse late-stage difluoromethylation of many pharmaceuticals and natural products were readily realized. Notably, a new difluoromethylated PD-1/PD-L1 immune checkpoint inhibitor was conveniently synthesized and showed both improved metabolic stability and enhanced antitumor efficacy. Preliminary mechanistic studies suggested the involvement of a Ni(I/III) catalytic cycle.

[1] Pharm-X Center, College of Pharmaceutical Sciences, Shanghai Jiao Tong University, 800 Dongchuan Road, Shanghai 200240, China. [2] These authors contributed equally: Heng Lu, Ruo-Xuan Xiao. ✉email: ao6919zhang@sjtu.edu.cn

In recent decades, fluorine-containing molecules have gained tremendous importance in the field of drug discovery, approximately accounting for over 30% of the total and crossing diverse disease indications[1–4]. Among these, gem-difluoromethylated ones are a unique class of organofluoro compounds, due to the capability of difluoromethyl motif both in creating new patentable intellectual property and in improving druglike properties, such as metabolically masking of carbonyl, bioisosteric replacement of methylene, amide, ether and esters[5,6]. More specifically, the aryldifluoromethyl aryl ether module (ArCF$_2$OAr') has attracted increasing interests recently due to its optimal metabolic stability against benzylic oxidation by CYP450 enzyme in liver and similar or slightly improved biological activity, compared to the non-fluorinated precursors ArCH$_2$OAr'[7–12]. A few representative druglike compounds bearing ArCF$_2$OAr' structural scaffold are listed in Fig. 1, including antitumor kinase inhibitors I and II[7,8], antimalarial compound III[9], guanylate cyclase activator IV[10], antivirus compound V[11], and PD-1/PD-L1 interaction inhibitor VI[12].

Compared to alkyldifluoromethyl aryl/alkyl ethers, approaches to access aryldifluoromethyl aryl ethers are limited. Early in 1990[13], Zupan and coworkers demonstrated a XeF$_2$-introduced fluorination-rearrangement reaction of diarylketones under strong acidic condition, leading to aryldifluoromethyl aryl ethers (Fig. 1b-a). Later, Sekiya[14] and Hitchcock[15] respectively reported some modifications of this strategy by using various moderate acids. However, this strategy was restricted by the use of tedious XeF$_2$ as fluorination agent and suffered from narrow substrate compatibility. The classical oxidative desulfurization-difluorination of esters using Lawson's reagent or alkyl dithiols is also an option of choices to prepare aryldifluoromethyl aryl ethers (Fig. 1b-b). This method also suffers from several disadvantages, such as using smelly reagents and unstable

fluorination reagents[16–18]. Direct nucleophilic substitution of aromatic gem-difluoromethyl halides or its equivalents by phenols has been reported to deliver aryldifluoromethyl aryl ethers readily (Fig. 1b-c-I)[19–21]. As an improvement, Hu's group recently presented a radical nucleophilic substitution strategy of fluoroalkyl sulfones substituted phenanthridines by phenolates to build ArCF$_2$OAr' module through a single-electron transfer (SET) process (Fig. 1b-c-II)[22]. Shortly thereafter, Young's group[23,24] developed a frustrated Lewis-pair-mediated C-F activation of trifluoromethylarenes with 2,4,6-triphenylpyridine (TPPy) leading to active ArCF$_2$-TPPy salts, which then underwent substitution with suitable lithium aryloxides to afford aryl aryloxydifluoromethyl ethers (Fig. 1b-c-III). Unfortunately, these nucleophilic substitution strategies suffer from harsh reaction conditions and limited substrate scopes. Recently, Qing and co-workers[25] developed a direct C-H aryloxydifluoromethylation of heteroarenes through Ag-catalyzed decarboxylation of aryloxydifluoroacetic acids to afford ArCF$_2$OAr' species (Fig. 1b-d-I). However, this approach is limited to heteroaryl compounds.

Despite these available approaches to access ArCF$_2$OAr', they are generally limited to harsh reaction conditions, complex fluorine agents, and narrow substrate scopes. Therefore, developing a mild, highly substrate compatible, and site-selective method for convenient construction of ArCF$_2$OAr' remains highly desirable. Inspired by the wide use of mono- or difluoromethyl halides in the transition-metal catalyzed C-C coupling reactions with different partners[26–37], herein, we reported a Ni-catalyzed Suzuki cross-coupling reaction by using aryloxydifluoromethyl bromides (ArOCF$_2$Br) as a unique halide species to undergo coupling with arylboronic acids, leading to highly efficient synthesis of various aryl aryloxydifluoromethyl ethers (Fig. 1b-d-II).

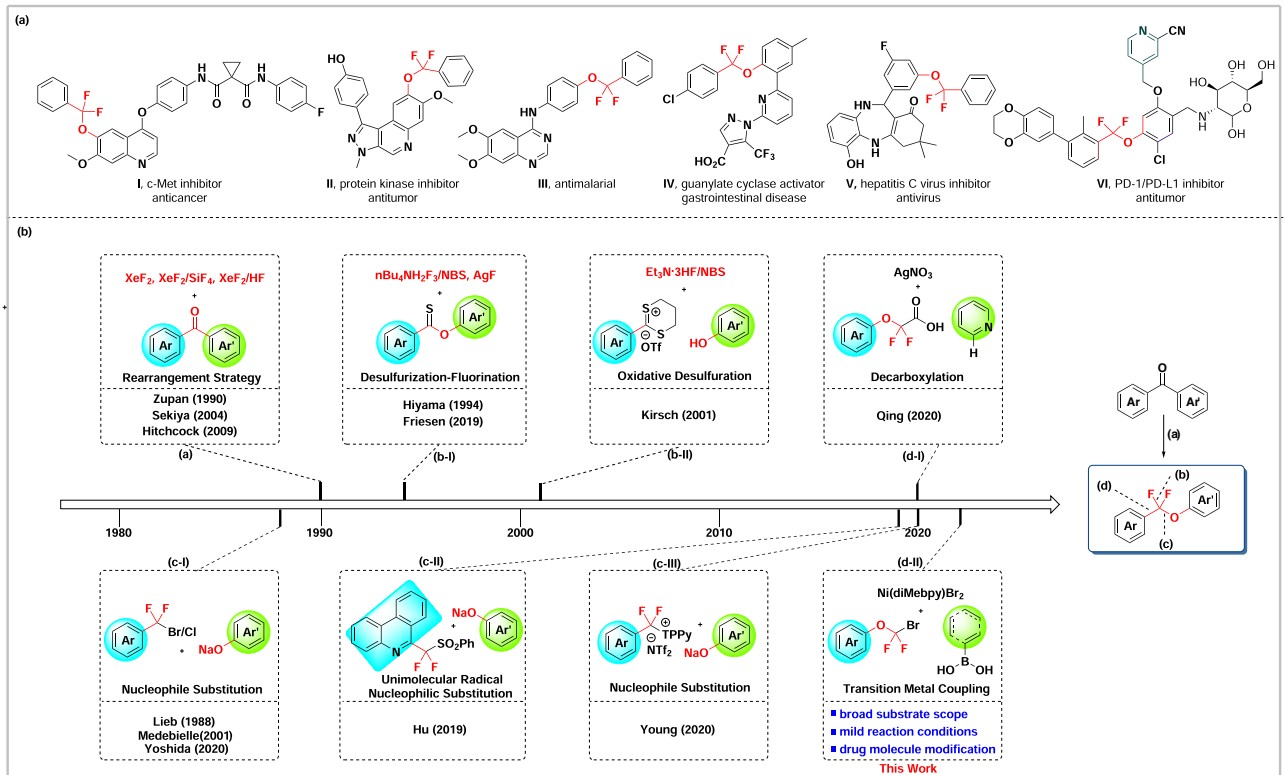

**Fig. 1 Overview of methods to prepare aryl aryloxydifluoromethyl ethers. a** Examples of biologically active aryl aryloxydifluoromethyl ethers. **b** Synthetic methods for aryl aryloxydifluoromethyl ethers.

**Table 1 Reaction optimization[a].**

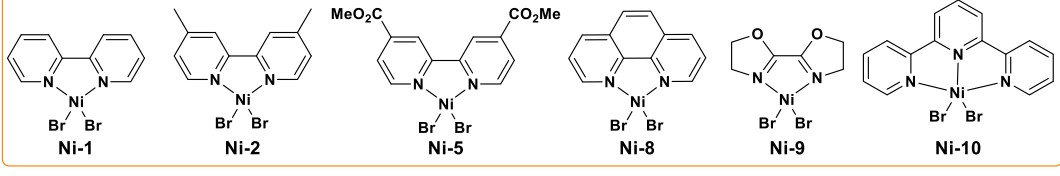

| Entry | NiBr$_2$L (10 mmol%) | Solvent | Additive (10 mmol %) | Yield[b] (3a/4/5, %) |
|---|---|---|---|---|
| 1 | Ni-1 | DMF | – | 21/15/27 |
| 2 | Ni-1 | NMP | – | N.D./N.D./N.D. |
| 3 | Ni-1 | DMSO | – | N.D./N.D./N.D. |
| 4 | Ni-1 | dioxane | – | N.D./N.D./37 |
| 5 | Ni-1 | acetone | – | 40/16/N.D. |
| 6 | Ni-1 | EA | – | 36/33/15 |
| 7 | Ni-1 | MeCN | – | 32/19/11 |
| 8[c] | Ni-1 | acetone | – | 45/10/N.D. |
| 9[d] | Ni-1 | acetone | – | N.D./5/14 |
| 10[c] | Ni-2 | acetone | – | 55/15/N.D. |
| 11[c] | Ni-5 | acetone | – | 38/21/N.D. |
| 12[c] | Ni-8 | acetone | – | 32/17/18 |
| 13[c] | Ni-9 | acetone | – | N.D./N.D./11 |
| 14[c] | Ni-10 | acetone | – | N.D./N.D./15 |
| 15[c] | Ni-2 | acetone | Py | 67/12/5 |
| 16[c] | Ni-2 | acetone | PPh$_3$ | 59/10/21 |
| 17[c] | Ni-2 | acetone | NPh$_3$ | 60/13/11 |
| 18[c] | Ni-2 | acetone | quinuclidine | 82/6/N.D. |
| 19[c] | Ni-2 | acetone | DABCO | 90(86)[e]/6/N.D. |
| 20[c] | – | acetone | DABCO | N.D./N.D./N.D. |

[a]Reaction condition: **1a** (0.4 mmol), **2a** (0.2 mmol), NiBr$_2$L (10 mol%), additive (10 mol%), K$_2$CO$_3$ (0.5 mmol), solvent (3 mL), 80 °C, 10 h.
[b]Yields were determined by GC-MS with n-dodecan as an internal standard.
[c]Using dry acerone as solvent.
[d]One drop of water was added.
[e]Isolated yield.

## Results and discussion

**Optimization studies.** Firstly, we used phenylboronic acid (**1a**) and (4-tolyloxy)difluoromethyl bromide (**2a**) as the model substrates, (2,2'-bipyridine)nickel(II) dibromide (Ni-1) as the catalyst, K$_2$CO$_3$ as the base and DMF as the solvent (for details see Table S1–Table S7 in SI). The expected product **3a** was obtained in 21% yield, along with oxodefluorinated **4** (15%)[38–40], hydrodebrominated **5** (27%)[41–44] and phenol (**6**, trace) as byproducts (Table 1, entry 1). Then, we tested different solvents to reduce the oxygendefluorination liability (entries 1-8) and acetone was found as a more appropriate solvent to afford product **3a** in 40% yield and byproducts **4** and **5** were much suppressed (entry 5). with the participation of water, **4** may result from Ni-CF$_2$OAr complex via oxidative defluorination or from the hydrolysis of product **3**. The yield of **3a** was further improved to 45% when dry acetone was used (Table 1, entry 8). As a contrast, adding water (one drop) to the reaction led to no detectable product (entry 9). Next, various Ni-containing catalysts were examined (entries 10-14). The Ni complex Ni-2 bearing an electronic-rich ligand showed higher efficiency than phen-Ni complex and other diamine or triamine bonding Ni complexes, providing **3a** in 55% yield (entry 10). Further, an additive was added to modulate the electronic and steric properties of the nickel center to facilitate the catalytic cycle[30–32]. Among the different nitrogen and phosphorus ligands tested, electron-donating alkylamines were found more effective than pyridine derivatives, arylamines, and phosphorus ligands. The best result was obtained using DABCO as the ligand, providing the expected product **3a** in 90% yield (entry 19). Ni-complexed catalyst was found necessary for this coupling (Table 1, entry 20).

**Scope of the reaction.** With the optimized reaction condition in hand (Table 1, entry 19), the scope of arylboronic acids was investigated. As shown in Fig. 2, various arylboronic acids bearing either electron-withdrawing or electron-donating substituents were successfully coupled with (4-tolyloxy)difluoromethyl bromide **2a** to deliver the corresponding aryloxydifluoromethylated products **3a-3z** in moderate to good yields. Arylboronic acids bearing a hydroxymethyl or cyano-substituent survived very well in this reaction and the corresponding products **3i** and **3o** were obtained in 75% and 55% yields, respectively, which provide potentials for further functional transformation. In certain cases, a mixed solvent of acetone/DMF was used to increase the

**Fig. 2 Ni-catalyzed aryloxydifluoromethylation reactions with diversified arylboronic acids. a** Reaction conditions: unless otherwise noted, a solution of 1 (0.2 mmol), 2a (0.4 mmol), Ni-2 (10 mol%), DABCO (10 mol%), and $K_2CO_3$ (0.5 mmol) in dry acetone (3.0 mL) was performed at 80 °C under argon for 10 h. The yields are isolated yields by column chromatography on silica gel. **b** Dry acetone (2.5 mL) and DMF (0.5 mL) were used for solvent. **c** 100 mg 4 A MS was added. **d** Dry acetone (1.5 mL) and DMF (1.5 mL) were used for solvent.

solubility of substrates and molecular sieves was added to suppress the defluorination liability of the products.

It is noted that arylboronic acids containing electron-donating functional groups at the *para*-position, such as methoxyl and methylthioyl, failed to give the expected difluorinated products, due to oxidative defluorination during purification by silica gel column chromatography. Slightly lower yields were obtained for arylboronic acids bearing ortho-substituents (**3v**-**3x**, 66–76%). Bicyclic and naphthyl boronic acids were found suitable as well to provide the corresponding products **3y** and **3z** in 79% and 69% yields, respectively. Meanwhile, cyclohex-1-en-1-ylboronic acid also successfully participated in this coupling reaction and provided the product **3aa** in 78% yield. To our delight, quinolinyl, pyridinyl, thiophenyl and dibenzothiophenyl boronic acids underwent the reaction smoothly as well leading to corresponding products **3ab**-**3ae** in 57-89% yields. We have also tried unsubstituted pyridine-boronic acids as substrates, and we failed to detect any corresponding products. More practically, this Ni-catalyzed reaction was also applied to arylboronic acids derived from biologically active compounds (including anti-inflammatory, anti-

bacterial, anti-hyperlipidemic drugs and glucose) and the corresponding products **3af**-**3aj** were readily obtained in moderate to good yields.

To further explore the substrate scope, various aryloxydifluoromethyl bromides were explored to cross-couple with arylboronic acids. As shown in Fig. 3, aryloxydifluoromethyl bromides bearing different substituents on the aryl including methoxy (**7a**), methyl (**7c**), and chloro (**7d**, **7g**) readily went through the coupling reactions with arylboronic acid and afforded corresponding products in high yields. Biphenyloxydifluoromethyl bromide bearing a hydroxymethyl substituent was also a suitable substrate for coupling with arylboronic acids yielding compounds **7e** and **7h** in 71% and 56% yields, respectively, which can be used for further transformation. Bicyclic aryloxydifluoromethyl bromides smoothly participated in the reaction as well giving compounds **7b** and **7f** in 71% and 57% yields, respectively. Notably, cross-coupling of arylthiodifluoromethyl bromide with arylboronic acids also occurred smoothly to deliver the corresponding difluoromethylated species **7i** and **7j** in 53% and 40% yields, respectively (Fig. 3).

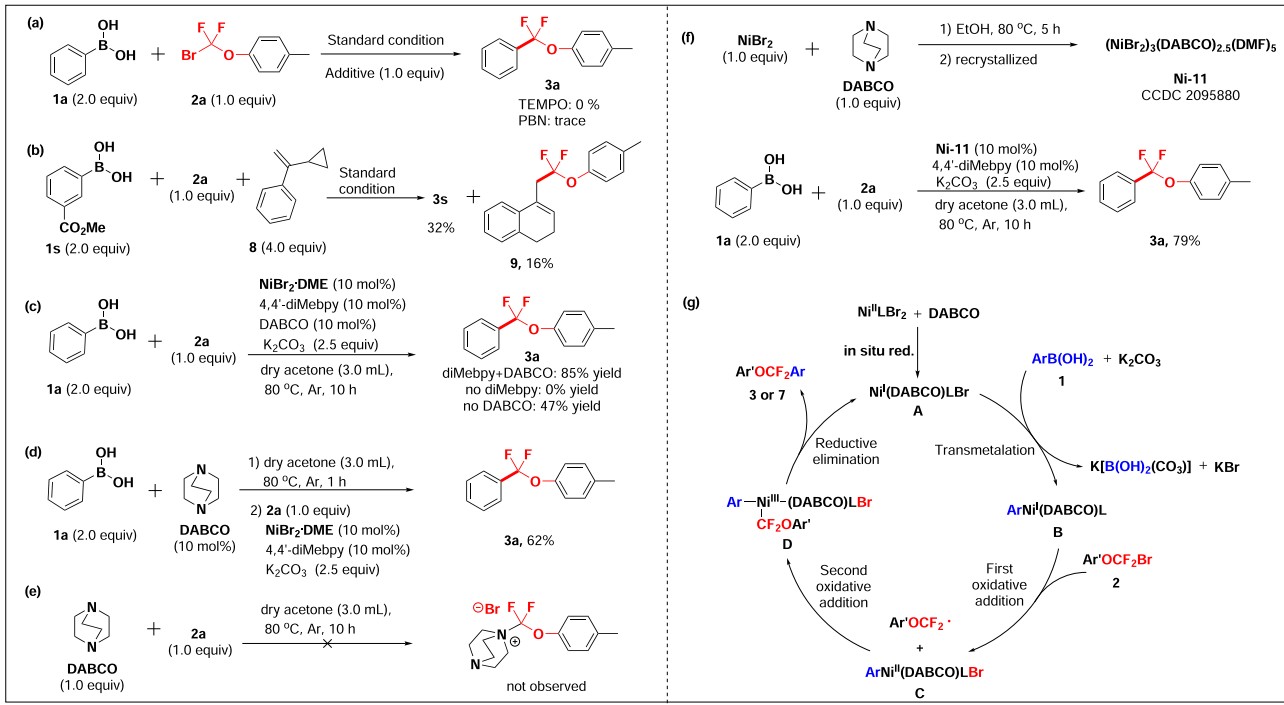

**Fig. 3 Ni-catalyzed aryloxydifluoromethylation reactions with various difluoromethyl bromides. a** Reaction conditions: unless otherwise noted, a solution of **1** (0.2 mmol), **2** (0.4 mmol), Ni-2 (10 mol%), DABCO (10 mol%), 100 mg 4 A MS and $K_2CO_3$ (0.5 mmol) in dry acetone (2.5 mL) and DMF (0.5 mL) was performed at 80 °C under argon for 10 h. The yields are isolated yields.

**Fig. 4 Mechanistic studies and proposed mechanism. a** Radical inhibition experiments. **b** Radical clock experiments. **c** The reaction proceeds under NiBr₂·DME, 4,4'-diMeby and DABCO. **d** The reaction proceeds through pretreatment of DABCO and phenylboronic acid **1a**. **e** Salt-forming experiment between DABCO and aryloxydifluoromethyl bromides **2a**. **f** The reaction proceeds under Ni(II) complexes[(NiBr₂)₃(DABCO)₂.₅(DMF)₅] and 4,4'-diMeby. **g** Proposed catalytic cycles.

**Mechanistic investigations**. To verify the reaction mechanism, several investigational experiments were conducted (Fig. 4). First, we found that the model reaction of **1a** and **2a** did not occur when the radical trapping reagent TEMPO (1.0 equiv) or N-*tert*-butyl-α-phenylnitrone (PBN, 1.0 equiv) was applied (Fig. 4a). Meanwhile, when the single electron transfer inhibitor 1,4-dinitrobenzene was added, the reaction was also blocked, indicating that a single-electron transfer of radicals was involved in this reaction (for details see SI, page S48). Moreover, a radical trapping product **9** (16% yield) was obtained when α-cyclopropyl styrene **8** was added in the reaction of **1s** and **2a** under standard condition (Fig. 4b).

In order to explore whether DABCO (1,4-diaza[2.2.2]bicyclooctane) acted as a co-ligand to coordinate with the nickel complex to promote the reaction, we conducted a few additional experiments. Firstly, the catalyst NiBr₂·DME, 4,4'-diMeby and DABCO were added to the reaction of **1a** and **2a** under the standard condition (Fig. 4c). Product **3a** was detected in 85% yield, which was nearly identical to the result obtained under optimum condition (90% yield). On the other hand, in the absence of DABCO, the yield of product was drastically reduced (55%); however, in the absence of the bipyridine ligand, the reaction failed to take place, which indicates that bipyridine ligand is an essential request and DABCO has a strong role in facilitating the reaction. Further, pretreating DABCO with phenylboronic acid **1a** in acetone at 80 °C for 1 h, followed by addition of substrate **2a**, NiBr₂·DME, 4,4'-diMeby and $K_2CO_3$ led to product **3a** in much reduced yield (62%, Fig. 4d). This result

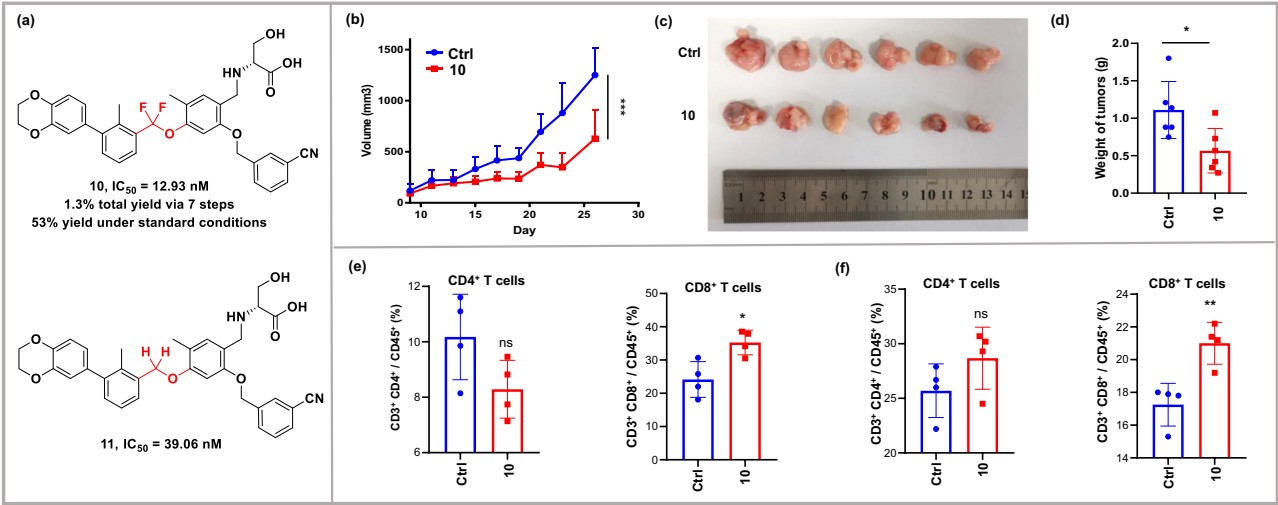

**Fig. 5 Treatment with compound 10 inhibited tumor growth in vivo and remodeled facilitated the infiltration of CD8+ T cells. a** PD-1/PD-L1 immune checkpoint inhibitor activity of difluorinated compound **10** and non-fluoro compound **11**. **b** The growth of transplanted MC38 tumors after local injection of 20 mg/kg compound **10** or vehicle daily. Data are presented as mean ± SEM ($n = 8$, *** $P < 0.001$, two-way ANOVA). **c** Image of excised tumors from. **d** weight of excised tumors from. **e** Representative plots (left) and frequency (right) of CD4+ T cells and CD8+ T cells in tumors. **f** Representative plots (left) and frequency (right) of CD4+ T cells and CD8+ T cells in spleens, data are presented as mean ± SD ($n = 4$, ns $P > 0.05$, *$P < 0.05$, **$P < 0.01$, $t$ test).

suggests that DABCO has no activating effect on the arylboronic acid. In addition, to exclude the possible activation of difluoromethyl bromide **2a** via DABCO to give an active electrophilic species as a reaction precursor, we treated these two species equivalently for 10 h at 80 °C, and no reaction precursor (a quaternary ammonium salt) was detected (analyzed by $^{19}$F NMR) (Fig. 4e). Finally, we synthesized nickel-DABCO complex [(NiBr$_2$)$_3$(DABCO)$_{2.5}$(DMF)$_5$] (**Ni-11**), which was characterized through X-ray single crystal diffraction (for details see Fig. S1 and Table S8 in SI). Compound **3a** was achieved in a comparable yield of 79% when phenylboronic acid **1a** was reacted with **2a** in the presence of the complex **Ni-11** and the challenging ligand 4,4'-diMeby (Fig. 4f). Together, all these experiments confirm that DABCO plays as a co-ligand to coordinate with the nickel catalyst and facilitates the catalytic cycle.

**Proposed mechanism.** Based on our experimental results and literature reports[45–50], we envisioned a tentative mechanism involving a Ni(I/III) catalytic cycle. As shown in Fig. 4g, a nickel (I) complex [Ni(I)(DABCO)LBr] (**A**) might be formed first in situ reduction of nickel (II) species through boronic acid, which then undergoes a transmetalation process with arylboronic acid **1** and K$_2$CO$_3$ to form arylnickel complex [ArNi(I)(DABCO)L] (**B**). Subsequent reaction with aryloxydifluoromethyl bromide **2** occurs likely through a single electron transfer pathway to produce the arylnickel(aryloxydifluoromethyl) complex [ArNi(II) (DABCO)LBr] (**C**) and aryloxydifluoromethyl radical. Further oxidative addition of aryloxydifluoromethyl radical to the complex **C** generates the key intermediate **D**. Finally, products **3** or **7** are obtained from a reductive elimination of the Ni(III) species **D**, along with regeneration of Ni(I) species **A** to complete the catalytic cycle.

**Difluoromethylated PD-1/PD-L1 immune checkpoint inhibitor.** To further confirm the practicability of the new Ni-catalyzed cross-coupling protocol, we applied this method to succefully prepare a new difluoromethylated PD-1/PD-L1 checkpoint inhibitor **10** in 1.3% total yield via 7 steps (for details see SI, page S39-S47). Diaryldifluoromethyl ether intermediate was obtained in 53% yield under standard conditions, which was difficult to prepare

using traditional oxidative desulfurization-difluorination method in our previous report (less than 10% yield)[12]. As shown in Fig. 5a, compared to the non-fluoro precedent **11**, difluorinated **10** showed three-fold higher potency against PD-1/PD-L1 interaction (IC$_{50}$ for **10/11**: 12.93 vs 39.06 nM). In vivo, inhibitor **10** displayed a significant reduction in tumor burden than control group with no significant loss of body weight or other common toxic effects in MC38 subcutaneous transplanted tumor model (Fig. 5b, c, d). Furthermore, to explore the effect of **10** on tumor immunity, we measured the percentage of T lymphocytes in tumors and spleens of mice treared with **10**. As shown in Figs. 5e, f (for details see Fig. S2 in SI), injection of **10** significantly increased the population of CD8+ T cells in both tumors and spleens, which indicated that **10** activated antitumor immune response in MC38 xenograft model.

## Conclusion

In summary, we have established an efficient strategy for the synthesis of a unique class of aryldifluoromethyl aryl ethers via Nickel-catalyzed cross-coupling of aryldifluoromethoxy bromides with arylboronic acids. The reaction showed wide substrate scope in both substrates containing various functional groups, and allowed late-stage difluoromethylation of many pharmaceuticals and natural products. Mechanistic studies revealed that this reaction might go through a Ni(I/III) catalytic cycle. Attractively, this method was successfully applied to readily synthesize a difluorinated PD-1/PD-L1 immune checkpoint inhibitor which showed much improved antitumor efficacy both in biochemical assay and in in vivo mice model. Therefore, this new protocol would provide ample potentials in the drug design and discovery filed.

## Methods
**Supplementary Methods.** For more details, see Supplementary Information page S1.

**General procedure for the synthesis of compound 3.** To a dried 10 ml Schlenk-type tube equipped with a magnetic stir bar was charged with arylboronic acid **1a** (49.0 mg, 0.4 mmol, 2.0 equiv), Ni-2 (8.0 mg, 0.02 mmol, 10 mol %), DABCO (2.3 mg, 0.02 mmol, 10 mol %) and K$_2$CO$_3$ (69.0 mg, 0.5 mmol, 2.5 equiv) under air. The reaction mixture was then evacuated and backfilled with Ar (3 times). 1-(bromodifluoromethoxy)-4-methylbenzene **2a** (47.0 mg, 0.2 mmol, 1.0 equiv), and

acetone (3 mL) were added. The mixture was stirred at 80 °C for 10 h. After cooled to room temperature, the reaction mixture was filtered and the filtrate was concentrated. The residue was purified on a preparative TLC with petroleum ether/ethyl acetate as the eluent to afford the products **3a**.

**General procedure for the synthesis of compound 7**. To a dried 10 ml Schlenk-type tube equipped with a magnetic stir bar was charged with arylboronic acid **1 s** (72.0 mg, 0.4 mmol, 2.0 equiv), Ni-2 (8.0 mg, 0.02 mmol, 10 mol %), DABCO (2.3 mg, 0.02 mmol, 10 mol %), 100 mg 4 Å MS and $K_2CO_3$ (69.0 mg, 0.5 mmol, 2.5 equiv) under air. The reaction mixture was then evacuated and backfilled with Ar (3 times). 1-(bromodifluoromethoxy)-4-methoxybenzene **2b** (50.4 mg, 0.2 mmol, 1.0 equiv), and acetone (2.5 mL) and DMF (0.5 mL) were added. The mixture was stirred at 80 °C for 10 h. After cooled to room temperature, the reaction mixture was filtered and the filtrate was concentrated. The residue was purified on a preparative TLC with petroleum ether/ethyl acetate as the eluent to afford the products **7a**.

**General procedure for the synthesis of metal-complex Ni-11**. To a stirring solution of DABCO (336 mg, 3.0 mmol, 1.0 equiv) in EtOH (30 mL) was added $NiBr_2$ (648 mg, 3.0 mmol, 1.0 equiv), the reaction mixture was stirred at 80 °C for another 5 h, then the solution was filtrated and the filtrate waswhich was recrystallized from DMF and ᵗBuOMe to give an orange solid.

**Compound characterization**. For more details, see Supplementary Data 1.

**In vivo effect of compound 10**. 6-week-old female mice were purchased from Shanghai SLAC Laboratory Animal Co.,Ltd. The animal experimental protocols were approved by the Instructional Animal Care and Use Committee of Shanghai Jiao Tong University. $1 \times 10^6$ MC38 cells were suspended in PBS and subcutaneously injected into the right axilla region of the mice. The volume of tumors were measured with vernier caliper and calculated using the formula: volume $(mm^3) = 0.5 \times$ longest diameter $\times$ shortest diameter. When tumor volumes reached 100 $mm^3$ (Day9), the mice were randomly divided into two groups, and subcutaneously injected with control solvent (5% DMSO, 55% PEG400, 40% $H_2O$) or compound **10** (20 mg/kg) around the tumor daily. Tumor volumes and body weight of mice were measured every two days. After 18 days, mice were scarified and the tumors were collected and weighted, and the spleens were collected simultaneously.

**Immune cells infiltration in tumors and spleens**. Tumor tissues were minced and digested in 1 mg/mL hyaluronidase, 1 mg/mL collagenase IV and 0.15 mg/mL DNase I for 2 hours and filtered with 70 μm strainer. Mononuclear cells were isolated with Ficoll-Paque PREMIUM 1.073 (Cytiva) as the manufacturer's instruction, and blocked with Purified Rat Anti-Mouse CD16/CD32 (BD Pharmingen) for 1 hours at 4 °C. After staining with Fixable Viability Stain 520, anti-mouse CD3e PerCP-Cy5.5, anti-mouse CD8a APC and anti-mouse CD4 PE (BD Pharmingen) for 30 min at 4 °C, the cells were washed and analyzed by a Cyto-FLEX flow cytometry (BECKMAN COULTER). The spleens were grinded and filtered with 70 μm strainer, followed by lysing red blood cell using BD Pharm Lyse™ lysing solution. The spleen cells were collected, blocked, stained and analyzed as described above. The data were analyzed by FlowJo 10.6.2.

**Reporting summary**. Further information on research design is available in the Nature Research Reporting Summary linked to this article.

## Data availability

All the relevant data of this study are available within this paper and its Supplementary Information are available from the corresponding author upon reasonable request. This includes Supplementary Data 1, which includes all NMR spectra. In addition, the X-ray crystallographic data for **Ni-11** are included in Supplementary Data 2 and Supplementary Data 3. Crystallographic data could be obtained free of charge from The Cambridge Crystallographic Data Center with the accession code CCDC 2095880 via www.ccdc.cam.ac.uk/data_request/cif.

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

## Acknowledgements

We thank the financial support from the National Innovation of Science and Technology-2030 (Program of Brain Science and Brain-Inspired Intelligence Technology) Grant of China (2021ZD0204004), the Major Projects for Shanghai Zhangjiang National Independent Innovation of China (ZJ2021-ZD-007), and the grants (WF540162618, AF1700037, WF220217002, WH101117001) from Shanghai Jiao Tong University.

## Author contributions

H.L. and R.X.X. contributed equally to this work. H.L. planned, conducted, analyzed and summarized the experiments. R.X.X. established MC38 xenograft model and performed the in vivo antitumor activity assay. C.Y.S., Z.L.S. and H.W.L. participate in the experiments or discussions. H.L. and C.Y.S. wrote the manuscript with feedback from all authors. A.Z. supervised the research. All authors approved the paper.

## Competing interests

The authors declare no competing interests.
