## [Peer review file · Communications Chemistry]

Reviewers' comments:

Reviewer #1 (Remarks to the Author):

The manuscript under review by Heng Lu et al. describes the development of a nickel catalysed Suzuki coupling between boronic acids and bromodifluoromethyl aryl ethers and thioethers. The report is a useful synthetic extension of known chemistry, and provides an alternative means to access known/accessible compounds. One of the main drawbacks to this chemistry would appear to be the preparation of the requisite bromodifluoromethyl group, which requires multiple steps (in the SI) and isn't as attractive as a selective coupling to a CF₃ group. Nonetheless, I believe that it will be suitable for publication in Communications Chemistry after a few minor revisions.

Revisions in the manuscript:

- The introduction provides a thorough outline of prior chemistry, however there are some missing reports that are relevant and should be included. For example, Suzuki couplings for difluorobenzyl fragments were disclosed by Zhang (Org. Chem. Front., 2015, 2, 38 and J. Am. Chem. Soc. 2021, 143, 13971) and Young (Org. Lett. 2021, 23, 1915), and should be included with references 24-30. Further, Yoshida has made difluorobenzyl thioethers directly (Chem. Eur. J. 2020, 26, 6136) and ethers indirectly (Org. Lett. 2020, 22, 9292). This is accomplished from the trifluoromethyl group, so is synthetically easier to access
- "oxygendifluorinated" should be oxodefluorinated
- "hydrogendetabrominated" should be hydrodebrominated
- "electron-rich" should be electron-donating
- Figure 4a "TMEPO" should be TEMPO
- The Figure 4a depicts 1a, while the text refers to 1s. Please clarify.
- Figure 4d: What is Ni-11? Include it in Figure 4
- Control reactions don't prove Ni-DABCO complex is formed or required
- Without spectrometric, spectroscopic or kinetic support for the role of DABCO as a ligand, I don't think that there is sufficient experimental evidence to claim it is a co-ligand.
- Mechanistic proposal is extremely tentative with many 'gaps'. How is the Ni(II) reduced to Ni(I)? Do the Ni(I) species have a halide ligand? If so, how is this regenerated during the catalytic cycle? What undergoes metathesis in the transmetallation step (halide?, carbonate?)?
- What is the yield of 10? Why not include this in the substrate scope above so the reaction conditions and yield are provided to the readers?
- Figure 5 (and the toxicity/inhibition study) seems a little ad-hoc in this synthetic methodology paper. It is encouraging to show application, but there is little explanation/discussion on what the toxicology results signify. Could Figure 5 be simplified if it is not going to be explained in depth?

Revisions in the SI:

- In the General Experimental: nm not nM
- Table S1 – there is no positive result. Why is the result with ideal conditions not included?
- What is the reasoning behind additive selection?
- Table S7: missing two OH

Reviewer #2 (Remarks to the Author):

As Zhang and his co-workers described in the submitted manuscript that aryl aryloxydifluoromethyl ethers is important bioactivity moiety in drug or druglike molecules, various methods have been developed by numerous chemists. Zhang exhibited a different pathway to synthesis of the aryl aryloxydifluoromethyl ethers through Nickel-catalyzed aryloxydifluoromethylation with aryl boronic acids. Despite a similar Nickel-catalyzed difluoroalkylation of aryl boronic acids has been depicted by X.G. ZHANG (Angew. Chem. Int. Ed. 2014, 53, 9909–9913) to construct difluoroalkylated arenes, this

present work still has its synthesis value to the special aryl aryloxydifluoromethyl ethers compounds.

In the first part, aryl aryloxydifluoromethyl ethers could be obtained in a mild condition with good tolerance of different functional groups. In the second part, encouraged by the success of the late-modification of many pharmaceuticals and natural products, the authors synthesized a difluorinated PD-1/PD-L1 immune checkpoint inhibitor. The further biochemical test showed the difluorinated inhibitor exhibited good bioactivity.

Considering the importance of the aryl aryloxydifluoromethyl ethers and the inspired exploiting perspective of the metal-catalysis organic synthesis, I support the manuscript publication in Communications Chemistry.

However, there are some problems about the manuscript listed below:

- 1> "eq." or "equiv." should be "equiv" in table and figure.
- 2> "10 mmol%" should be "10 mol%" in table 1.
- 3> "0 %" should be "N.D." in table 1.
- 4> The "(a)", "[a]" and "[a]" in text should be consistent.
- 5> "As a contrast, adding more water" should be "As a contrast, adding water" in optimization part.
- 6> "Tetrahedron Letters." should be "Tetrahedron Lett." in reference 15.
- 7> "Jones, G. D. et al." in reference 39 should not appropriate. All authors should be listed.

Reviewer #3 (Remarks to the Author):

The work in the paper is of interest to others in the community as a novel method to access challenging aryl difluoromethyl aryl ethers is exemplified. Given the relatively mild conditions and the use of the broad aryl boronic acid monomer set, the reaction has potential to be of real use to the broader Organic community. A few key questions from a review of the presented work:

1. The authors discuss optimisation of the reaction and present their data in Table 1. The side product 4 is notable and it would be good if the authors could comment further on its formation. Compounds of this type have potential chemical instability issues under certain conditions (e.g. acidic) and the formation of the side product could be an artefact of not having a basic additive present (an obvious question is how would trialkylamine bases work as additives?). Commentary on the stability of all products made and which are the most likely to be unstable would be useful here - the compounds won't have viability as drug motifs if they are prone to hydrolysis under mild conditions.
2. Commentary on whether a boronate ester is a viable coupling partner would also be useful - these are often more available and stable than boronic acids so good to quantify here.
3. For figure 2, the heterocyclic scope is relatively limited. It would be good to see an unsubstituted 3-pyridyl coronate as a coupling partner as these are often very challenging and if the reaction works here it would be significant.
4. For figure 3, the scope of the aryl on 2 seems very limited - where are the het examples here and are these challenging to access/use?
5. The proposed mechanism seems reasonable and the data presented reasonable.

Overall, this is a strong paper which is a significant contribution to the field. Addressing the questions raised above and broadening the scope of the examples would increase likelihood that the chemistry would have applicability in a pharmaceutical industry setting.

This reviewer is supportive of publication if these comments are addressed.

Reviewer 1

(1) The manuscript under review by Heng Lu et al. describes the development of a nickel catalysed Suzuki coupling between boronic acids and bromodifluoromethyl aryl ethers and thioethers. The report is a useful synthetic extension of known chemistry, and provides an alternative means to access known/accessible compounds. One of the main drawbacks to this chemistry would appear to be the preparation of the requisite bromodifluoromethyl group, which requires multiple steps (in the SI) and isn't as attractive as a selective coupling to a CF₃ group.

Our response: Thanks. As commented, the protocol is indeed an efficient synthesis to difluorinated product, however, the preparation of the requisite bromodifluoromethyl substrates needs several steps. More convenient bromodifluoromethyl sources are being exploited in our laboratory.

(2) The introduction provides a thorough outline of prior chemistry, however there are some missing reports that are relevant and should be included. For example, Suzuki couplings for difluorobenzyl fragments were disclosed by Zhang (Org. Chem. Front., 2015, 2, 38 and J. Am. Chem. Soc. 2021, 143, 13971) and Young (Org. Lett. 2021, 23, 1915), and should be included with references 24-30. Further, Yoshida has made difluorobenzyl thioethers directly (Chem. Eur. J. 2020, 26, 6136) and ethers indirectly (Org. Lett. 2020, 22, 9292). This is accomplished from the trifluoromethyl group, so is synthetically easier to access.

Our response: Thanks for the suggestion. We have added the above references in the revised manuscript.

(3) “oxygendefluorinted” should be “oxodefluorinated”.

Our response: Thanks. We have corrected this error.

(4) “hydrogende brominated” should be “hydrode brominated”.

Our response: Thanks. We have corrected this error.

(5) “hydrogende brominated” should be “hydrode brominated”.

Our response: Thanks. We have changed “hydrogende brominated” to “hydrode brominated”.

(6) “electron-rich” should be “electron-donating”.

Our response: Thanks for this, and we have corrected this error.

(7) Figure 4a “TMEPO” should be “TEMPO”.

Our response: Thanks. We have corrected this error.

(8) The Figure 4a depicts **1a**, while the text refers to **1s**. Please clarify.

Our response: Thanks. We have changed “**1s**” to “**1a**” in the text.

(9) Figure 4d: What is **Ni-11**? Include it in Figure 4.

Our response: Thanks for this, **Ni-11** is a metal complex $[(\text{NiBr}_2)_3(\text{DABCO})_{2.5}(\text{DMF})_5]$, which synthesized from NiBr_2 and DABCO (for details see the SI). We have added it in Figure 4.

(10) Control reactions don't prove Ni-DABCO complex is formed or required.

Our response: Thanks. We believe that Ni-DABCO complex is formed in this reaction by the following experiments. First, we synthesized nickel-DABCO complex $[(\text{NiBr}_2)_3(\text{DABCO})_{2.5}(\text{DMF})_5]$ (**Ni-11**), which was characterized through X-ray single crystal diffraction (for details see the SI). Then, when catalyzed by **Ni-11** and bipyridine ligand, the yield of product (79%) was comparable to the yield (86%) under standard condition (nickel-bipyridine ligand complex and DABCO). On the other hand, when DABCO was lacked, the yield of product was drastically reduced (55%). In addition, if

bipyridine ligand was lacked, the reaction couldn't take place. The above experiments suggest that DABCO should be involved in the formation of nickel-DABCO complex as a co-ligand. Finally, we had attempted to cultivate single-crystal cultures of the three-component complexes (NiBr₂, 4,4'-diMeppy and DABCO), unfortunately, it was not successful. We have revised the mechanism discussion to cover these details.

(11) Without spectrometric, spectroscopic or kinetic support for the role of DABCO as a ligand, I don't think that there is sufficient experimental evidence to claim it is a co-ligand.

Our response: Thanks. Using an additive to promote Ni-catalyzed Suzuki cross-coupling reaction is a widely reported strategy. For example, Zhang's group studied the role of DMAP in Ni-catalyzed Suzuki cross-coupling reaction (Angew. Chem. Int. Ed. 2015, 54, 9079; Nat. Commun., 2018, 9, 1170.). Based on the above reports, we believe that there are three possible roles for DABCO to facilitate this reaction. The first possible role is the activation of the arylboronic acids to facilitate the transmetalation. To verify this, we pretreated DABCO with phenylboronic acid **1a** in acetone at 80 °C for 1 h, then adding substrate **2a**, NiBr₂·DME, 4,4'-diMeppy and K₂CO₃ led to product **3a** in 62% yield, which is lower than that lacking pre-treatment condition (85% yield). This result doesn't support that DABCO has an activating effect of the arylboronic acids. The second possible role is the activation of the difluoromethyl bromides to give an active electrophilic species as a reaction precursor. However, when we treated these two species equivalently for 10 h at 80 °C, and no reaction precursor (a quaternary ammonium salt) was detected through ¹⁹F NMR. Therefore, this possibility should be excluded. The last possible role is that DABCO might serve as a co-ligand to facilitate the catalytic cycle. To verify this, we synthesized a Ni-DABCO complex [(NiBr₂)₃(DABCO)_{2.5}(DMF)₅] (**Ni-11**), and used **Ni-11** and 4,4'-diMeppy as catalysts led to product **3a** of 79% yield, which was comparable to the yield (85%) under the condition using NiBr₂·DME, 4,4'-diMeppy and DABCO as catalysts. On the other hand, the yield dropped to 47% without DABCO and the reaction failed to take place without 4,4'-diMeppy. Therefore, we believe that DABCO likely serves as a co-ligand to

facilitate the catalytic cycle. We have revised the mechanism discussion to cover these details.

(12) Mechanistic proposal is extremely tentative with many gaps. How is the Ni(II) reduced to Ni(I)? Do the Ni(I) species have a halide ligand? If so, how is this regenerated during the catalytic cycle? What undergoes metathesis in the transmetallation step (halide?, carbonate?)?

Our response: Thanks for this comment. We have modified the mechanism discussion in Figure 4. In situ reduction of nickel (II) species through boric acid has been reported (Chem. Soc. Rev., 2013, 42, 5270; Trends Chem., 2019, 1, 830.). Nickel (I) species have a bromide anion as the ligand. The bromide anion joins with boric acid via transmetallation and then leaves the catalytic cycle. The bromide anion subsequently enters the catalytic cycle from aryloxydifluoromethyl bromides **2** via oxidative addition. In the presence of potassium carbonate, Ni (I) species and boric acid undergo transmetallation step.

(13) What is the yield of **10**? Why not include this in the substrate scope above so the reaction conditions and yield are provided to the readers?

Our response: Thanks. Compound **10** was obtained in 1.3% total yield via 7 steps, diaryldifluoromethyl ether intermediate was obtained in 53% yield under standard conditions. We have added the corresponding yields in Figure 5.

(14) Figure 5 (and the toxicity/inhibition study) seems a little ad-hoc in this synthetic methodology paper. It is encouraging to show application, but there is little explanation/discussion on what the toxicology results signify. Could Figure 5 be simplified if it is not going to be explained in depth?

Our response: Thanks. The pharmacological experiments in Figure 5 were performed to prove that the novel Ni-catalyzed cross-coupling approach could be used in the development of anti-tumor drugs, and the resulting compounds indeed presented the

ability to activate anti-tumor immune response. To make the structure of the article more reasonable, we simplified Figure 5 as your suggestion.

(15) Revisions in the SI: In the General Experimental: nm not nM

Our response: Thanks. We have changed this item in the SI.

(16) Table S1: there is no positive result. Why is the result with ideal conditions not included?

Our response: Thanks. In Table S1, we focus on comparing the application of different transition metal-catalyzed difluoroalkylation coupling reaction conditions to aryloxydifluoromethyl bromides and boronic acids. We have added the ideal conditions to Table S1.

(17) What is the reasoning behind additive selection?

Our response: Thanks. We want to use an additive as a co-ligand to modulate the electronic and steric properties of the nickel center to facilitate the catalytic cycle, and some additives with lone pairs of electrons were tried, including pyridine derivatives, aryl phosphines, aryl amines and alkyl amines.

(18) Table S7: missing two OH

Our response: Thanks for this, and we have corrected this error.

Reviewer 2

(1) As Zhang and his co-workers described in the submitted manuscript that aryl aryloxydifluoromethyl ethers is important bioactivity moiety in drug or druglike molecules, various methods have been developed by numerous chemists. Zhang exhibited a different pathway to synthesis of the aryl aryloxydifluoromethyl ethers through Nickel-catalyzed aryloxydifluoromethylation with aryl boronic acids. Despite a similar Nickel-catalyzed difluoroalkylation of aryl boronic acids has been depicted by X.G. ZHANG (Angew. Chem. Int. Ed. 2014, 53, 9909-9913) to construct

difluoroalkylated arenes, this present work still has its synthesis value to the special aryl aryloxydifluoromethyl ethers compounds.

Considering the importance of the aryl aryloxydifluoromethyl ethers and the inspired exploiting perspective of the metal-catalysis organic synthesis, I support the manuscript publication in Communications Chemistry.

Our response: Thank you for your positive comment.

(2) “eq.” or “equiv.” should be “equiv” in table and figure.

Our response: Thanks for this, and we have corrected this error.

(3) “0 %” should be “N.D.” in table 1.

Our response: Thanks. We have changed “0 %” to “N.D.” in table 1.

(4) The (a) and [a] in text should be consistent.

Our response: Thanks. We have corrected this error.

(5) “As a contrast, adding more water” should be “As a contrast, adding water” in optimization part.

Our response: Thanks. We have corrected this error.

(6) “Tetrahedron Letters.” should be “Tetrahedron Lett.” in reference 15.

Our response: Thanks. We have corrected this error.

(7) Jones, G. D. et al. in reference 39 should not appropriate. All authors should be listed.

Our response: Thanks. We have listed all authors.

Reviewer 3

(1) The work in the paper is of interest to others in the community as a novel method to access challenging aryl difluoromethyl aryl ethers is exemplified. Given the relatively mild conditions and the use of the broad aryl boronic acid monomer set, the reaction has potential to be of real use to the broader Organic community. A few key questions from a review of the presented work:

The authors discuss optimisation of the reaction and present their data in Table 1. The side product **4** is notable and it would be good if the authors could comment further on its formation. Compounds of this type have potential chemical instability issues under certain conditions (e.g. acidic) and the formation of the side product could be an artefact of not having a basic additive present (an obvious question is how would trialkylamine bases work as additives?). Commentary on the stability of all products made and which are the most likely to be unstable would be useful here - the compounds won't have viability as drug motifs if they are prone to hydrolysis under mild conditions.

Our response: Thanks. By-product **4** may result from the defluorinated oxidation of the difluoromethylether-nickel complex or from the hydrolysis of product **3** with the participation of water, corresponding explanation has been added to the text.

Based on mechanistic experiments as well as literature reports, we propose that DABCO serve as a co-ligand to facilitate the catalytic cycle. On the other hand, the assistance provided by 10 mol% DABCO is limited compared to the basic environment created by 2.5 equivalents of K_2CO_3 .

We have discussed the stability issue of the difluorinated compounds by presenting a few products containing electron-donating functional groups at the para-position, such as methoxyl and methylthioly, that were unstable and decomposed during purification by silica gel column chromatography due to oxidative defluorination. Otherwise, the products are quite stable. Indeed, as an unusual pharmacological replacement of methoxy moiety, difluoromethyleneoxy is a valuable structural motif, which has gained more and more application in medicinal chemistry either to improve drug-like properties. The metabolic stability of these compounds has been confirmed in the preparation of PD-L1 inhibitors recently by our group (J. Med. Chem. 2021, 64, 16687).

(2) Commentary on whether a boronate ester is a viable coupling partner would also be useful - these are often more available and stable than boronic acids so good to quantify here.

Our response: Thanks. We screened different aryl boronic acid derivatives and only aryl boronic acid gave good yields, please refer to SI, Table S7 for details.

(3) For figure 2, the heterocyclic scope is relatively limited. It would be good to see an unsubstituted 3-pyridyl boronate as a coupling partner as these are often very challenging and if the reaction works here it would be significant.

Our response: Thanks. This is a good suggestion and we have tried unsubstituted 3-pyridyl boronic acid and 4-pyridyl boronic acid in our substrate screening, unfortunately, we failed to detect any corresponding products.

(4) For figure 3, the scope of the aryl on 2 seems very limited - where are the het examples here and are these challenging to access/use?

Our response: Thanks. The screening of the range of substrates is mainly to determine the scope of application of the reaction. In Figure 2, we screened the substrates for a variety of functional groups and also included some heterocyclic and drug molecules for application, indicating good compatibility of the reaction. Therefore, in Figure 3, we have only performed a partial screening of functional groups and drug molecules. In addition, synthesis of bromodifluoromethyl ethers requires a decarboxylative bromination process in which some of the functional groups are not compatible.

(5) The proposed mechanism seems reasonable and the data presented reasonable. Overall, this is a strong paper which is a significant contribution to the field. Addressing the questions raised above and broadening the scope of the examples would increase likelihood that the chemistry would have applicability in a pharmaceutical industry setting. This reviewer is supportive of publication if these comments are addressed.

Our response: Thank. We have addressed the concerns and added more discussion in the revised manuscript.

REVIEWERS' COMMENTS:

Reviewer #1 (Remarks to the Author):

Most of the changes that the authors have made satisfy the concerns/comments that I had made. However, I still feel that their catalytic cycle is extremely tentative. Why do they need a carbonate base if they are proposing direct metathesis between the aryl boronic acid and Ni-Br species? Oxo bases are usually added because an anion exchange (between a halide and oxo base) occurs and the metathesis step with the aryl boronic acid and oxo base is kinetically and thermodynamically favored (as compared to metathesis with a M-Br species). What is the role of the carbonate base in the proposed cycle if the by-product is $\text{BrB}(\text{OH})_2$? Is it simply a drying agent? If so, it is strange that such large effects are seen by varying the oxo base. I would suggest either not specifying the boronic by-product if it is not known, or including $\text{K}[\text{B}(\text{OH})_2(\text{CO}_3)]$ and KBr as the by-products. Further, a control reaction of heating DABCO with the boronic acid prior to addition of other reagents is not going to be productive, as the heating was done in acetone, which will undergo a base induced aldol condensation to release water (which is why it is very difficult to get very dry acetone) and the boronic acid will form boroxine and also release water. Addition of water was shown to be detrimental by the authors, so presumably heating the boronic acid in acetone prior to the reaction would also be detrimental. However, based on precedence, the role of DABCO as a co-ligand is accepted. The manuscript has no further small errors, however, the SI seems to be not the final version. There are author comments and tracked changes still visible. Additionally, under the synthesis of Ni-11, please remove the typo "waswhich" and "n-dodecan" should be "n-dodecane" throughout the SI.

Reviewer #3 (Remarks to the Author):

Appropriate changes have been made and I would support publication.

Reviewer 1

(1) Most of the changes that the authors have made satisfy the concerns/comments that I had made. However, I still feel that their catalytic cycle is extremely tentative. Why do they need a carbonate base if they are proposing direct metathesis between the aryl boronic acid and Ni-Br species? Oxo bases are usually added because an anion exchange (between a halide and oxo base) occurs and the metathesis step with the aryl boronic acid and oxo base is kinetically and thermodynamically favored (as compared to metathesis with a M-Br species). What is the role of the carbonate base in the proposed cycle if the by-product is BrB(OH)_2 ? Is it simply a drying agent? If so, it is strange that such large effects are seen by varying the oxo base. I would suggest either not specifying the boronic by-product if it is not known, or including $\text{K[B(OH)}_2\text{(CO}_3\text{)]}$ and KBr as the by-products.

Our response: Thanks. We agree with this reviewer on the discussion and suggestion on the transmetalation process mechanism. We agree that K_2CO_3 plays a role as a facilitator of anion exchange in transmetalation process. We didn't identify the by-products in transmetalation process and agree with the suggestion on $\text{K[B(OH)}_2\text{(CO}_3\text{)]}$ and KBr as the by-products, and would seek alternative ways to track them in the future. We have modified the discussion in the revised TEXT.

(2) Further, a control reaction of heating DABCO with the boronic acid prior to addition of other reagents is not going to be productive, as the heating was done in acetone, which will undergo a base induced aldol condensation to release water (which is why it is very difficult to get very dry acetone) and the boronic acid will form boroxine and also release water. Addition of water was shown to be detrimental by the authors, so presumably heating the boronic acid in acetone prior to the reaction would also be detrimental. However, based on precedence, the role of DABCO as a co-ligand is

accepted.

Our response: Thanks for this suggestion. We agree with the reviewer that under the promotion of DABCO, the aldol reaction of acetone and the polymerisation reaction of phenylboronic acid may produce water, and affect the yield of the reaction. However, we did not include this analysis in the discussion since we did not have straight evidence on this process.

(3) The manuscript has no further small errors, however, the SI seems to be not the final version. There are author comments and tracked changed still visible. Additionally, under the synthesis of Ni-11, please remove the typo “waswhch” and “n-dodecan” should be “n-dodecane” throughout the SI.

Our response: Thanks. We have put the SI into final revision. We have removed “waswhch”, and we have changed “n-dodecan” to “n-dodecane” in SI.

Reviewer 3

(1) Appropriate changes have been made and I would support publication.

Our response: Thanks for your positive comment.